# Establishing the effects of mesoporous silica nanoparticle properties on in vivo disposition using imaging-based pharmacokinetics

Prashant Dogra[1], Natalie L. Adolphi[2], Zhihui Wang [1,3], Yu-Shen Lin[4], Kimberly S. Butler[5,6,7], Paul N. Durfee [5,8], Jonas G. Croissant [5,6], Achraf Noureddine [5,6], Eric N. Coker [9], Elaine L. Bearer[10], Vittorio Cristini[1,3] & C. Jeffrey Brinker[5,6,8,11]

The progress of nanoparticle (NP)-based drug delivery has been hindered by an inability to establish structure-activity relationships in vivo. Here, using stable, monosized, radiolabeled, mesoporous silica nanoparticles (MSNs), we apply an integrated SPECT/CT imaging and mathematical modeling approach to understand the combined effects of MSN size, surface chemistry and routes of administration on biodistribution and clearance kinetics in healthy rats. We show that increased particle size from ~32- to ~142-nm results in a monotonic decrease in systemic bioavailability, irrespective of route of administration, with corresponding accumulation in liver and spleen. Cationic MSNs with surface exposed amines (PEI) have reduced circulation, compared to MSNs of identical size and charge but with shielded amines (QA), due to rapid sequestration into liver and spleen. However, QA show greater total excretion than PEI and their size-matched neutral counterparts (TMS). Overall, we provide important predictive functional correlations to support the rational design of nanomedicines.

[1] Mathematics in Medicine Program, Houston Methodist Research Institute, Houston, TX 77030, USA. [2] Department of Biochemistry and Molecular Biology, University of New Mexico, Albuquerque, NM 87131, USA. [3] Department of Imaging Physics, University of Texas MD Anderson Cancer Center, Houston, TX 78701, USA. [4] Department of Internal Medicine, University of New Mexico, Albuquerque, NM 87131, USA. [5] Center for Micro-Engineered Materials, University of New Mexico, Albuquerque, NM 87131, USA. [6] Chemical and Biological Engineering, University of New Mexico, Albuquerque, NM 87131, USA. [7] Sandia National Laboratories, Department of Nanobiology, Albuquerque, NM 87123, USA. [8] Cancer Research and Treatment Center, Molecular Genetics and Microbiology, University of New Mexico, Albuquerque, NM 87131, USA. [9] Sandia National Laboratories, Applied Optical and Plasma Science, Albuquerque, NM 87185, USA. [10] Department of Pathology, University of New Mexico, Albuquerque, NM 87131, USA. [11] Sandia National Laboratories, Self-Assembled Materials Department, Albuquerque, NM 87185, USA. These authors contributed equally: Prashant Dogra, Natalie L. Adolphi, Zhihui Wang. Correspondence and requests for materials should be addressed to V.C. (email: vcristini@houstonmethodist.org) or to C.J.B. (email: cjbrink@sandia.gov)

The implementation of nanotechnology in medicine promises to advance drug delivery and diagnostic imaging. Nanoparticle (NP)-based drug delivery and imaging systems, termed nanocarriers, have the potential to package and protect cargos that are too toxic, fragile, insoluble, or unstable to be delivered as free drugs or imaging agents. Nanocarriers can be engineered to package combined therapeutic and diagnostic cargos (the so-called theranostics) and equipped with a variety of triggering mechanisms to release cargo on demand according to intracellular or extracellular environmental stimuli. Further, it is possible to engineer the nanocarrier size, shape, and surface chemistry to enhance circulation times and direct the biodistribution of the drug or imaging agent within the organism by "passive" targeting, for example, by the enhanced permeability and retention (EPR) effect, wherein NPs passively accumulate in the tumor microenvironment due to its leaky vasculature characterized by fenestrations ~200–2000 nm in diameter[1]. Finally, by surface modification of the nanocarrier with targeting ligands that bind to receptors/antigens over-expressed on the cells of interest, it is possible to achieve precise administration of therapeutic cargos to specific cells or tissues via "active" targeting, while sparing collateral damage to healthy cells and potentially overcoming multiple drug resistance mechanisms[2].

Despite the established preclinical potential of nanocarriers as effective drug delivery vehicles and imaging agents, NP-based delivery has achieved only moderate success in clinical translation, especially for therapeutic nanomedicines. According to a comprehensive review surveying the literature from the past 10 years, the in vivo tumor delivery efficiency of nanocarriers, which has relied primarily upon the EPR effect, has averaged around only 0.7% of the injected dose[3]. This has been attributed to uncontrolled, non-specific interactions of NPs with the immune and microanatomical components of non-tumor sites, particularly the mononuclear phagocytic system (MPS) organs, namely liver, spleen, and bone marrow, that serve as "sinks" for preferential NP accumulation[4]. This is highly problematic as the clinical translation of nanotherapeutics demands a predetermined and reproducible disposition (biodistribution and clearance) profile of NPs needed to achieve the requirements of efficacy and safety. For instance, the US Food and Drug Administration (FDA) guidelines require that diagnostic agents be completely cleared from the body in a reasonable timeframe to avoid interference with other xenobiotics[5]. In contrast, it is particularly desirable to have prolonged systemic circulation of chemotherapy-loaded NPs for maximal exposure to tumor tissue and accumulation by the EPR effect[6]. Literature stipulates that a hydrodynamic size of under 5.5 nm and a positive zeta potential promote rapid renal clearance of NPs, which is ideal for diagnostic applications[5,7], but also that solid NPs exceeding 6 nm in diameter cannot be effectively renally cleared[5], occasionally shown to be untrue (vide infra). For therapeutic applications, such as cancer nanotherapy, polymeric coatings, such as polyethylene glycol (PEG) that serve to reduce serum protein adsorption (opsonization) on the NP surface, are proclaimed to enhance the longevity of NPs in circulation, ideal for increased exposure to the tumor[6,8], but so far the tumor-targeting efficiency of largely PEGylated NPs has been modest and highly variable[3].

We contend that the deficiencies of NP therapeutics and the confusion in the literature as to their efficiencies and behaviors are largely attributable to insufficient control of NP synthesis and the lack of in vivo colloidal stability, which have led to inconsistent biodistribution and have therefore prevented the establishment of definitive structure–activity relationships (SAR) necessary for successful preclinical development and clinical translation of nanocarriers. To date, based on the ten-year survey of NP delivery to solid tumors[3], several trends have been observed

with respect to NP physicochemical properties: inorganic NPs have higher delivery efficiencies than organic NPs, NPs smaller than 100 nm in hydrodynamic diameter have higher delivery efficiencies than larger particles, nearly neutrally charged NPs (defined as having zeta potentials −10 to +10 mV) have higher delivery efficiencies than more positively or negatively charged particles, and rod-shaped particles are more efficient than spherical or plate-like particles. These trends presumably reflect the in vivo stabilities of the NPs, differential uptake by the MPS, and differences in renal clearance; however, this survey did not establish unambiguously the stability or size polydispersity of the NPs nor their biodistribution, and there appeared to be no systematic studies to isolate the effects of size or charge or surface chemistry for NPs of comparable composition and shape. Previous biodistribution studies have shown that NP physicochemical properties, primarily size, charge, and surface polymeric coatings[9–12], along with routes of administration[13–15] are critical in governing the disposition kinetics of NPs, but again systematic comparisons are often lacking. Noteworthy in this regard, we have recently demonstrated for mesoporous silica NPs (MSNs) of identical size and charge that the spatial arrangement and accessibility of charged molecules on the MSN surface (i.e. surface chemistry) is another critical, but to date unrecognized, factor governing biological behavior of NPs[16].

Herein to establish quantitative SAR in vivo, we employ single photon emission computed tomography integrated with computed tomography (SPECT/CT) imaging of indium ($^{111}$In) radiolabeled, colloidally and compositionally stable, monosized, cargoless MSNs to determine biodistribution and clearance in healthy rats. By systematically varying MSN physicochemical variables in the therapeutically relevant size range of ~25–150 nm (corresponding to diameters of ~32–142 nm in our study), we examine the effect of size, zeta potential, and surface chemistry on in vivo disposition of hydrodynamically stable, non-targeted MSNs administered via intravenous (i.v.) or intraperitoneal (i.p.) injection. We employ SPECT/CT imaging to determine the disposition kinetics of MSNs within ten regions of interest (ROI) in the rat. We then develop a parsimonious, semi-mechanistic mathematical model to describe the macroscopic concentration–time behavior of MSNs in individual ROIs and estimate relevant pharmacokinetic (PK) parameters. Our results allow the formulation of significant correlations between MSN size and surface chemistry and PK parameters, thus enabling quantitative comparison of the disposition behavior of MSNs necessary to advance their status toward clinical use. An interplay between physiological and NP physicochemical variables governs the in vivo behavior of NPs, and this study furthers our understanding of this interaction.

## Results

**Characterization of MSNs**. The establishment of NP SAR in vivo demands consummate control of NP size, shape, surface chemistry, and stability. Here, to avoid confounding effects of particle size polydispersity and hydrodynamic instability (which have obscured the role of particle size in previous studies), we employed well-characterized, monosized (defined as hydrodynamic diameter polydispersity index (PdI) <0.1), PEGylated MSNs that exhibited long-term stability in physiologically relevant media (see Methods for details of MSN synthesis and characterization). $^{111}$In-labeled MSNs with three different surface chemistries were synthesized with nominal diameters of 50 nm: (1) PEG-polyethylenimine (PEG-PEI), (2) PEG-quaternary amine (PEG-QA), and (3) PEG-trimethylsilane (PEG-TMS) (see Fig. 1a). Additionally, PEG-TMS MSNs were synthesized with nominal sizes: 25, 90, and 150 nm. The MSN core diameter was

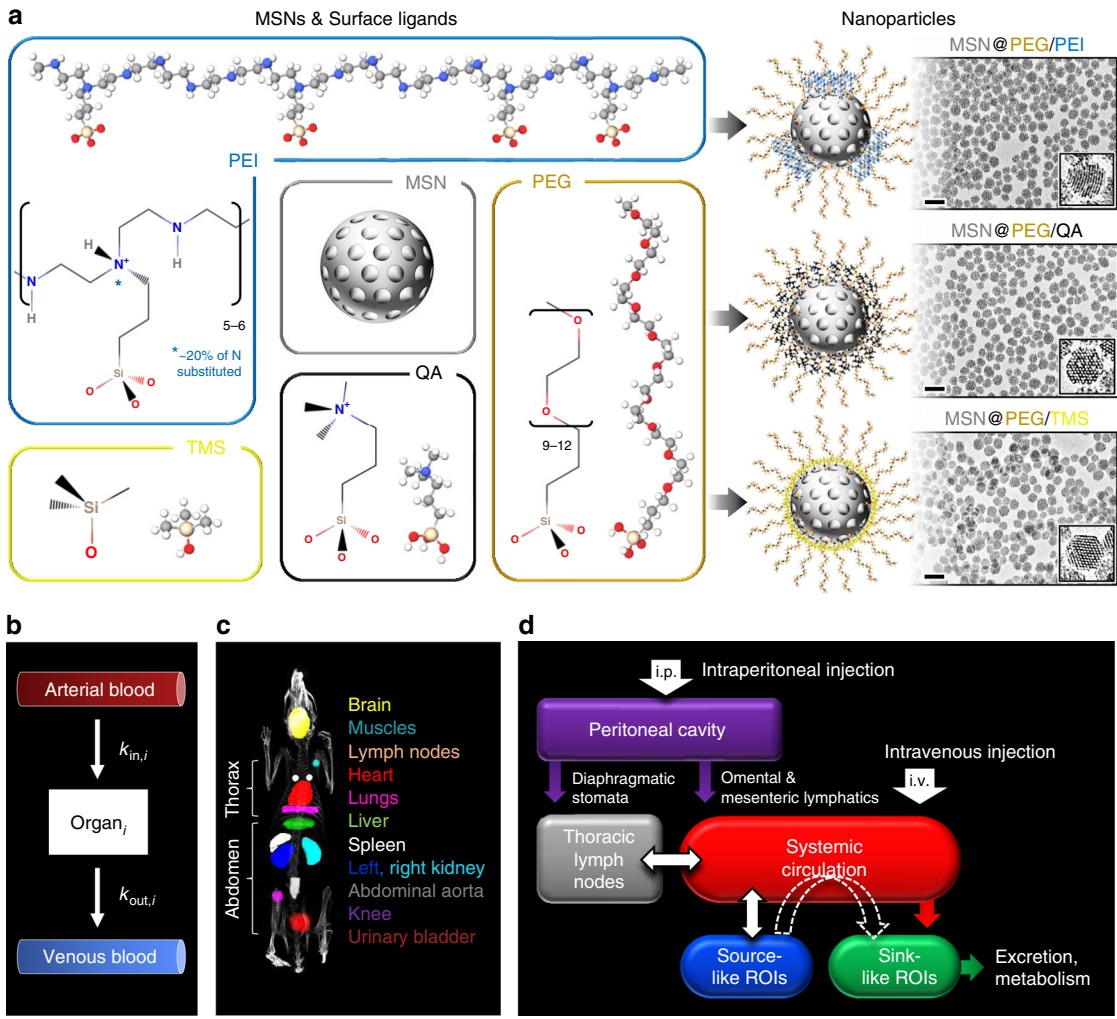

**Fig. 1** Design of MSNs, SPECT/CT imaging, and mathematical modeling. **a** Molecular models of surface ligands and the resulting MSNs used in the study, characterized by TEM. Polyethylenimine (PEI) is "patchy" and may extend beyond the polyethylene glycol (PEG) layer and cover the MSN surface intermittently, unlike the smaller quaternary amine (QA) and trimethylsilane (TMS) groups that remain shielded within the PEG layer and cover the MSN surface uniformly. PEI and QA groups provide a strongly positive zeta potential (ζ) to MSNs, while TMS makes them neutral. Scale bars: 100 nm. **b** A schematic of the underlying modeling hypothesis depicts an organ $i$ that receives influx of NPs from its major feeding artery, which after crossing the vasculature of organ $i$ exit into venous blood. Assuming the influx and efflux processes to both follow first-order kinetics with rate constants $k_{in,i}$ and $k_{out,i}$, respectively, we obtain a double-exponential function (Eq. (4)) to describe the concentration–time course of NPs in individual ROIs. **c** Regions of interest (ROIs) generated using inviCRO's Multi Atlas Segmentation Tool to perform quantification of whole-body radioactivity concentration. **d** Representation of the whole-body framework to understand the disposition of NPs. I.p. administration, unlike i.v. injection, is associated with absorption of NPs from the peritoneal cavity into systemic circulation through bowel lymphatics, causing accumulation of NPs in thoracic lymph nodes. Once in the systemic circulation through either route of injection, NPs are distributed across all organs in the body in proportion to organ blood flow rates. Once inside the organ microvasculature, NPs encounter traps that sequester NPs from circulation into the interstitial space. Based on the low or high density of traps, we can classify the organs into "source-like" and "sink-like," respectively. The former do not sequester NPs due to lack or low density of traps, unlike the latter, which generally trap NPs unless the physicochemical properties of NPs are unfavorable for entrapment. By allowing NPs to pass through their vasculature without sequestration, source-like organs thus become a secondary source of NPs for the sink-like organs (as depicted through the dotted white arrow), which eventually dispose of the NPs through metabolic and excretory pathways

determined by transmission electron microscopy (TEM) and the hydrodynamic diameter and PdI were determined by dynamic light scattering (DLS) (see Table 1 and Supplementary Fig. 1). Hydrodynamic diameters were consistently between 10 and 20 nm larger than core diameters determined by TEM, consistent with previous observations in the literature[17]. For all three surface chemistries the average pore diameter determined from nitrogen ($N_2$) adsorption isotherms using Non-Local Density Functional Theory assuming a silica surface and cylindrical pores was 3.5–3.8 nm (see Supplementary Fig. 2 and Supplementary Table 2). The uniformity and completeness of the surface

modification was assessed by measurement of the hemolytic activity of the various MSNs towards human red blood cells. It is well documented that amorphous, monosized colloidal silica NPs, for example, the so-called Stöber silica NPs, exhibit a very significant dose-dependent hemolytic potential[18] due to electrostatic and hydrogen bonding interactions of surface silanols (≡Si-OH and deprotonated silanols (≡Si-O⁻)) and siloxanes (≡Si-O-Si≡) with RBC membrane constituents[19]. Introduction of meso-porosity reduces necessarily the surface concentrations of silanols and siloxanes and accordingly reduces the hemolytic potential, but there remains significant hemolytic activity[18]. By comparison,

**Table 1 MSN characterization (size, polydispersity index, and zeta potential)**

| MSN ID | Surface coating | TEM diameter (nm) | DLS hydrodynamic diameter (nm) | Polydispersity index | Zeta potential (mV) |
|---|---|---|---|---|---|
| TMS25 | PEG-TMS | 32 ± 1 | 46 ± 0 | 0.068 | −5 ± 1 |
| TMS50 | PEG-TMS | 55 ± 1 | 69 ± 0 | 0.028 | −7 ± 1 |
| TMS90 | PEG-TMS | 93 ± 1 | 113 ± 1 | 0.022 | −7 ± 1 |
| TMS150 | PEG-TMS | 142 ± 1 | 162 ± 1 | 0.025 | −4 ± 0 |
| PEI50 | PEG-PEI | 52 ± 2 | 65 ± 0 | 0.030 | +37 ± 1 |
| QA50 | PEG-QA | 56 ± 2 | 66 ± 1 | 0.038 | +38 ± 2 |

Data represent mean ± s.d., $n = 3$. Refer to Supplementary Fig. 1 for TEM images of MSNs and Supplementary Fig. 4 for hydrodynamic stability of MSNs

the various PEG-TMS-modified, PEG-QA-modified, and PEG-PEI-modified MSNs studied here have essentially zero hemolytic activity (see Supplementary Fig. 3), meaning that the surface Si-OH groups are completely passivated by the PEG-TMS, PEG-QA, and PEG-PEI surface modifications. All three MSN types showed excellent hydrodynamic stability in 1× phosphate-buffered saline (PBS) over 5 to 7 days (Supplementary Fig. 4) where hydrodynamic diameter varied by <4.5%. Furthermore, crucial to the establishment of quantitative biodistribution data, we showed the indium-labeled MSNs to have excellent compositional stability. For all three surface chemistries, indium leaching studies conducted in simulated body fluid (SBF) showed <0.001% indium loss over 48 h (see Supplementary Fig. 5 and Methods for details). Surface charge of the particles was determined by measurement of zeta potential ($\zeta$). PEG-TMS-modified particles were nearly neutrally charged ($\zeta = -4$ to $-7$ mV), while PEG-PEI-modified and PEG-QA-modified particles were strongly positively charged with statistically identical zeta potential values ($\zeta = +37$ to $+38$ mV) (Table 1). In fact, PEI-modified and QA-modified MSNs are essentially indistinguishable according to the standard determinants of biodistribution (core size, hydrodynamic size, shape and zeta potential); however, as we will show, they varied greatly in their disposition (vide infra) due to differing distributions and exposures of surface amines (see Fig. 1a), consistent with previous observations of their ex ovo behaviors within a highly vascularized chorioallantoic membrane model[16].

**Generalized biodistribution of MSNs.** At the very outset, it is important to understand that the radioactivity observed in SPECT/CT images (see Fig. 2) potentially has two origins: (1) radioactivity from NPs circulating through the vasculature of an ROI, and (2) radioactivity from NPs sequestered in the extravascular space of an ROI. The former NPs are still bioavailable for delivery to a potential target site, but the latter are generally not, unless the organ they are sequestered in is the target organ itself. We propose that the extent of NP sequestration in the extravascular space of an ROI is dependent on the density of traps in the microvasculature of the ROI. Traps here are referred to as the three recognized microscopic mechanisms that work to remove NPs from circulation: (i) opsonization by plasma proteins[20], which label the NPs as foreign invaders for targeted phagocytosis[21], (ii) binding of NPs to vascular endothelial surfaces, which may lead to cellular internalization[22], and (iii) fenestrated capillaries and sinusoids allowing extravasation[1] of NPs into tissue interstitia or directing excretion[23]. Because these traps are not uniformly distributed across the body, rather are localized in higher densities in the MPS organs, we can classify the organs in the body according to high or low density of the most relevant physiological traps (phagocytes, fenestrae, interendothelial gaps)[24,25] as: (1) sink-like and (2) source-like organs, respectively. NPs in the "sink-like" organs can passively accumulate over time in the extravascular space, due to high activity of traps, and

eventually metabolized or excreted, leading to a permanent loss of bioavailable NPs. In contrast, NPs in source-like ROIs travel through the vasculature without getting trapped[26]. Thus, source-like ROIs collectively represent the blood pool through which NPs circulate and remain bioavailable for delivery to the target site or to sink-like organs.

From the representative SPECT/CT images in Fig. 2 and their quantification in Fig. 3, we can understand the generalized biodistribution behavior of MSNs and thus infer similarities and differences between groups. As seen following i.v. injection in Figs. 2a–c, g–i and 3a and Supplementary Fig. 6, at the 30-min time point, an almost identical concentration is observed across all groups in the thoracic region (heart and lungs). The exception however is PEI50 (Figs. 2i, 3a), where a much weaker signal is observed in the thorax. As seen in the quantified concentration–time data of heart (Fig. 3a), a significant difference is not observed between groups at 30 min (one-way analysis of variance (ANOVA) and Tukey's test, $P > 0.05$), except with PEI50. Over time, the concentration in the thorax tends to decline, although at different rates, suggesting that the organs in the thorax tend not to accumulate MSNs into their interstitium, and NPs are cleared from the blood pool of thoracic organs in a particle-type-dependent fashion. This justifies the classification of heart and lungs as source-like organs. In contrast, the concentration in the abdomen (spleen and liver) tends to rise to a maximum followed by a slow or zero decline within 24 h post injection; note that this behavior is also particle-type-dependent (see Figs. 2a–c, g–i and 3a and Supplementary Fig. 6). The rise of MSN concentration in MPS organs for prolonged periods of time suggests that MSNs tend to accumulate over time in the interstitium of these organs, hence a very small washout is observed within 24 h, justifying these organs to be classified as sink-like organs. The literature shows that over time the spleen and liver gradually clear the NP load through the hepatobiliary route of elimination and not through recirculation into blood[12,13,27,28]. Further, PEI50 MSNs (Figs. 2i, 3a) that exhibit the lowest concentration in heart and lungs at 30 min among all groups accordingly exhibit the highest accumulation in the liver and spleen at 30 min, indicating a rapid hepatic and splenic uptake of PEI50 MSNs from blood. As seen in Figs. 2a–c, g–i and 3a and Supplementary Fig. 6, the behavior of kidneys and urinary bladder appears to be consistent across groups, except for QA50 (Figs. 2h, 3a), where the bladder shows significantly larger activity over time than other groups. Also, in Fig. 2h QA50 shows radioactivity in the large intestine at the 5 h and 24 h time points, unlike other MSNs. Thus, it can be inferred that because of rapid urinary and fecal excretion, QA50 shows one of the lowest accumulations in the spleen and liver relative to other MSNs (see Fig. 3a). Finally, other ROIs, including abdominal aorta, brain, joints, and muscles exhibit only trivial concentrations (<1.5%ID g$^{-1}$ (percentage of injected dose per gram of tissue)) across groups. Their behavior, except joints, resembles that of source-like ROIs,

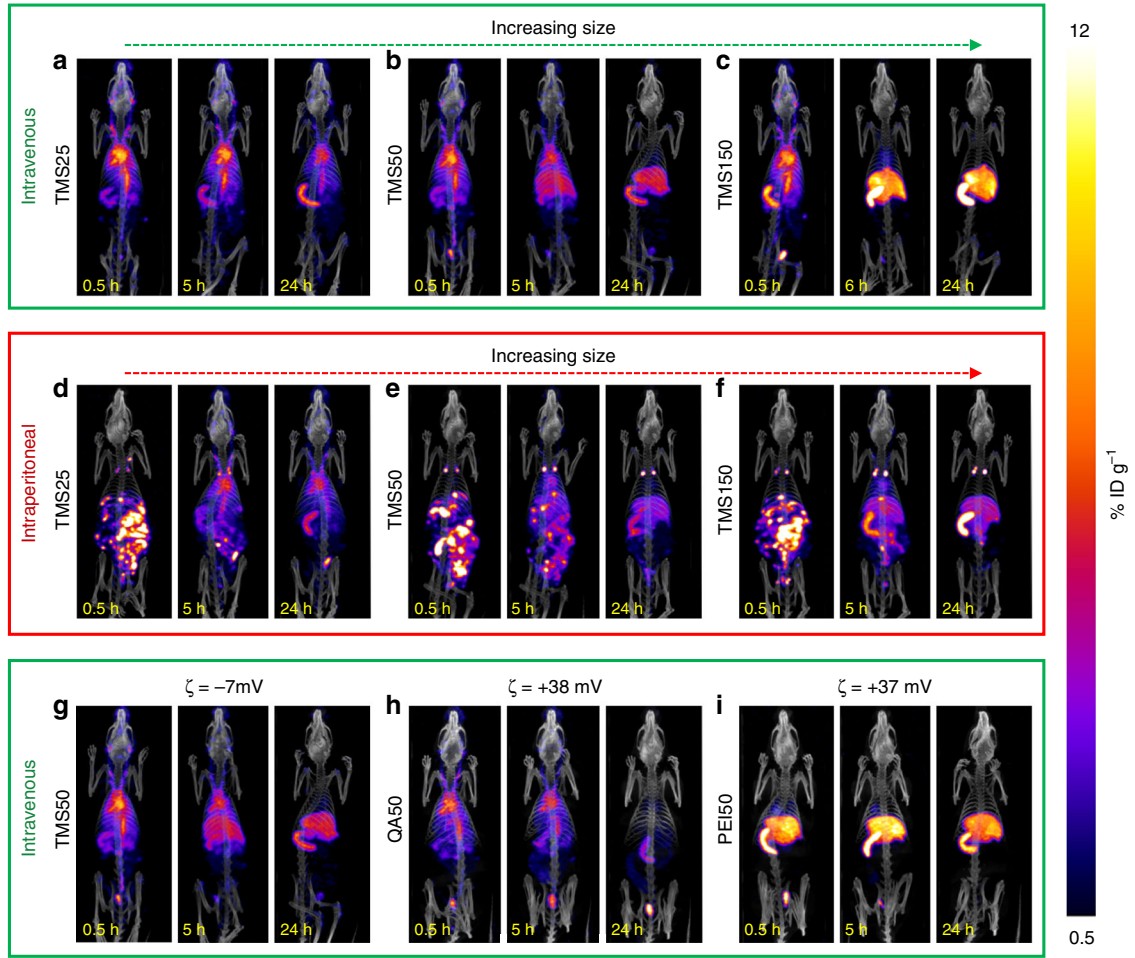

**Fig. 2** SPECT/CT images showing the whole-body spatio-temporal evolution of MSNs. PEG-TMS-coated MSNs of nominal sizes 25 nm (**a**, **d**), 50 nm (**b**, **e**), 90 nm (Supplementary Fig. 6), and 150 nm (**c**, **f**) were injected via i.v. (**a**–**c**, Supplementary Fig. 6) or i.p. (**d**–**f**) route to understand the effect of MSN size and route of administration. PEG-PEI-coated and PEG-QA-coated MSNs, namely, QA50 (**h**) and PEI50 (**i**), were administered i.v. to explore the effect of surface chemistry. Note that (**b**) and (**g**) are identical images (shown twice for ease of comparison). Also, TMS50 (**b**) and QA50 (**h**) were compared to understand the effect of zeta potential on disposition of MSNs. Injections were followed by SPECT/CT imaging at 30 min, 5 h (6 h in case of TMS150 (i.v.)), and 24 h. All SPECT images were scaled from 0.5 to 12%ID g$^{-1}$. Note: TMS90 MSN was not injected i.p.; lymph nodes and abdominal aorta were not analyzed as ROIs in the i.v. and i.p. cases, respectively

that is, a particle-type-dependent decline in concentration over time (see Fig. 3a, Supplementary Figs. 12a, b, 13a, b, 14a, b, and 15a, b).

Next, as seen in Fig. 2d–f, i.p. injection of PEG-TMS-coated MSNs shows a punctate biodistribution pattern throughout the abdomen at the initial time point of 30 min that seems to map the abdominal lymph circulatory network[29], with mediastinal lymph nodes in the thorax (see ROI map Fig. 1c) being an important site of radioactivity (see Figs. 2d–f, 3b). This initial phase represents the absorption of MSNs from the peritoneal cavity into blood[29,30]. Over time, however, the distribution pattern starts to resemble that of the corresponding i.v. cases for the three particle types (Fig. 2a–c), indicating that the MSNs have entered the systemic circulation. This behavior demonstrates the in vivo stability of the MSNs with respect to non-specific binding in lymph nodes. Having entered the circulatory system, i.p.-injected MSNs ultimately exhibit a mass transfer phenomenon similar to the one following i.v. administration, namely transfer of MSNs over time from source-like organs (e.g., heart and lungs) to sink-like organs (e.g., liver and spleen), and finally excretion (as depicted in Fig. 1d). The kinetics of these processes are, however, particle-type-dependent and in the subsequent sections we will unravel the effects of MSN physicochemical properties on the

kinetics of MSN disposition in blood, visceral organs, and excretory organs.

**Systemic kinetics**. We employed the SPECT-derived radioactivity concentration–time data of the heart ROI as a substitute for plasma concentration–time data[31–33] (assuming that MSNs in the heart ROI were in circulation due to its source-like character, i.e., lack of fenestrations large enough to allow NP escape from circulation[34,35]) to understand systemic kinetics of MSNs and estimate relevant PK parameters (see Figs. 3, 4a, b). Since the concentrations of different MSNs seem to vary in a mono-exponential or double-exponential fashion, we fit a one-compartment PK model[36] (Eq. (6) for i.v. and Eq. (4) for i.p.) to the concentration–time data and estimate model parameters (see Supplementary Tables 4 and 5). Fitted concentration–time curves demonstrate the effect of MSN size and route of administration (Fig. 4a) and surface chemistry and zeta potential (Fig. 4b) on systemic kinetics of MSNs. In Supplementary Fig. 7, normalizing the predicted concentration over time ($C(t)$) of individual MSNs by their own predicted concentration maxima ($C_{max}$) allows a direct comparison of different groups. We then estimated area under the curves (AUC$_{0-24\,h}$), uptake ($k_{in}$) and

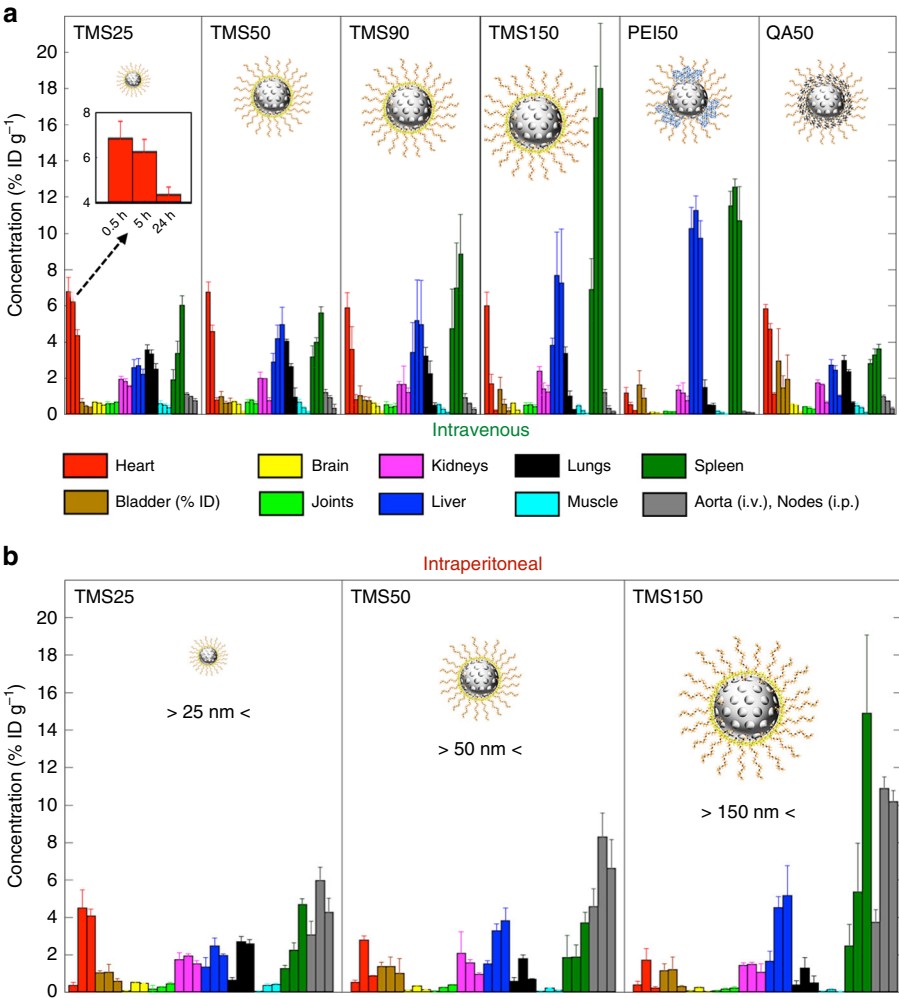

**Fig. 3** Whole-body quantitative biodistribution of MSNs. Bar plots of mean concentration–time data for various ROIs following **a**, i.v and **b**, i.p. injection of MSNs are shown. For each MSN-type, concentration (%ID g$^{-1}$) of MSNs in ten ROIs is shown at 30 min, 5 h (6 h for TMS150 (i.v.)), and 24 h, represented by three adjacent bars of same color (see inset in **a**). Data represents mean ± s.d., $n = 4$ (except TMS50 (i.p.) and TMS25 (i.p.), where $n = 3$). Note: Data for urinary bladder ROI is presented as activity (%ID) and not as concentration

elimination rate constants ($k_{out}$), and half-lives ($t_{1/2}$) of individual curves from Fig. 4a, b.

As seen in Fig. 4c, AUC$_{0–24 h}$ decreases monotonically with an increase in particle size in the studied size range of ~32 to ~142 nm, irrespective of the route of delivery, governed by the mathematical relation: AUC$_{0–24 h} = \lambda \cdot$ size$^{-n}$, where $\lambda$ and $n$ are fitted coefficients, and size refers to the core diameter of NPs. The value of the power coefficient is ~1 for both i.v. and i.p. cases, suggesting a strongly negative linear dependence (see Supplementary Fig. 19e). Further, the elimination rate constant ($k_{out}$) increases (and thus $t_{1/2}$ decreases) with an increase in size (see Fig. 4e and Table 2); however, one-way ANOVA reveals no significant difference in the uptake rate constant ($k_{in}$) values across i.p. administered cases ($P > 0.05$) (see Fig. 4d). This suggests that absorption of NPs from peritoneal cavity in the studied size range is independent of particle size[37] and that the systemic bioavailability through either route of administration is primarily a function of the $k_{out}$ parameter. Published hemodynamic studies[38–40] show that smaller particles tend to have smaller margination[38] probabilities in blood capillaries, in addition to being shielded by erythrocytes, thus escaping near-wall accumulation, resulting in reduced extravasation through fenestrations and reduced internalization by endothelium or

near-wall phagocytes[41]. Thus, greater protection from the traps in microvasculature yields a higher systemic bioavailability for smaller-sized particles.

Given that the i.v. injected MSNs are 100% bioavailable, the bioavailability fraction (calculated as the ratio of dose normalized AUC$_{0–24 h}$ of i.p. to AUC$_{0–24 h}$ of i.v.) of i.p. administered TMS25, TMS50, and TMS150 MSNs is 72.8, 66.6, and 79.6%, respectively, thus quantifying the incomplete absorption of MSNs from the peritoneal cavity into blood. It is however important to note that $t_{1/2}$ is not significantly different (unpaired $t$ test, $P > 0.05$) between corresponding MSNs injected through the i.v. and i.p. route (see Table 2), indicating that upon entering the blood stream MSN kinetics is independent of their route of administration, which again highlights their in vivo stability required for clinical translation.

Next looking at the effect of surface chemistry in Fig. 4b, c, e and Table 2, PEI50 (which has surface-exposed amines) has a ~9-fold lesser AUC$_{0–24 h}$ (unpaired $t$ test, $P < 0.0001$) and half the $t_{1/2}$ (unpaired $t$ test, $P < 0.05$) relative to size-matched and zeta potential-matched QA50 (which has obstructed surface amines). These results are consistent with our previously published report[16], where we demonstrated the difference in cellular and tissue interactions of PEG-PEI-coated and PEG-QA-coated MSNs in vitro and ex ovo in the highly vascularized chicken

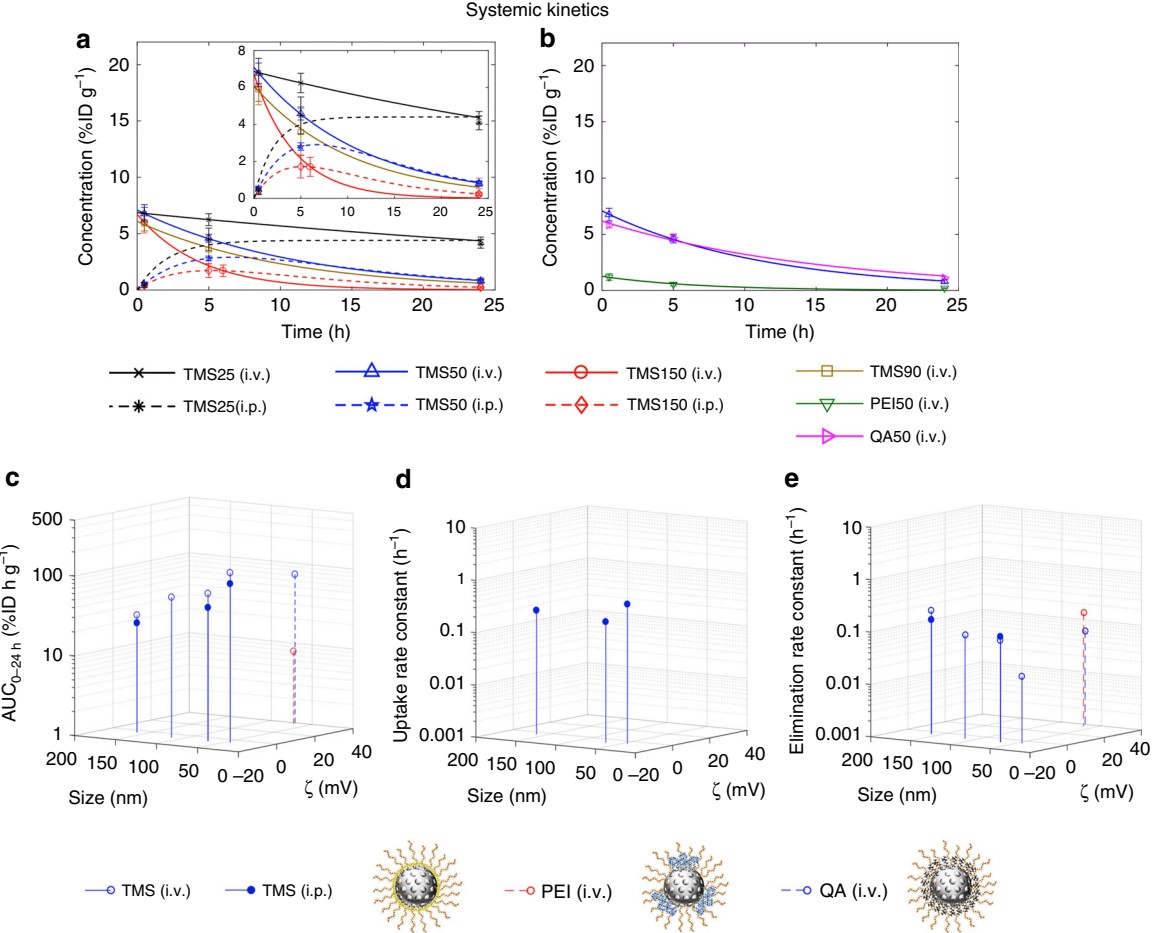

**Fig. 4** Systemic kinetics. **a**, **b** One-compartment PK model (Eq. (6) for i.v. and Eq. (4) for i.p.) was fit to the concentration–time data for different MSNs in the heart ROI (Supplementary Tables 4 and 5). Fitted concentration–time curves demonstrate the effect of MSN size and route of administration for TMS-modified MSNs (**a**) and surface chemistry and zeta potential for 50 nm diameter MSNs modified with TMS, QA, or PEI (**b**). The inset in **a** is a rescaled version of the figure for a clearer view. Solid lines, i.v. cases; dotted lines, i.p. cases. **c**–**e** 3-D stem plots show area under the concentration–time curves (AUC$_{0-24\,h}$) (**c**) and model parameter estimates ((**d**) uptake rate constant, $k_{in}$, and (**e**) elimination rate constant, $k_{out}$), obtained for different MSNs from **a**, **b**, in multiparameter space. Data represent mean ± s.d., $n = 4$ (except TMS50 (i.p.) and TMS25 (i.p.), where $n = 3$)

| Table 2 Estimated half-lives ($t_{1/2}$) for various MSNs | |
|---|---|
| **MSN ID** | **Half-life (h)** |
| TMS25 (i.v.) | 45.09 ± 17.70 |
| TMS25 (i.p.) | ~TMS25 (i.v.) |
| TMS50 (i.v.) | 7.83 ± 1.77 |
| TMS50 (i.p.) | 7.48 ± 2.06 |
| TMS90 (i.v.) | 6.87 ± 2.45 |
| TMS150 (i.v.) | 3.19 ± 1.07 |
| TMS150 (i.p.) | 4.97 ± 1.09 |
| PEI50 (i.v.) | 5.48 ± 3.24 |
| QA50 (i.v.) | 10.63 ± 0.22 |

Data represent mean ± s.d., $n = 4$ (except TMS50 (i.p.) and TMS25 (i.p.), where $n = 3$). Note: for TMS25 (i.p.), $t_{1/2}$ is not available because an elimination phase was absent in its concentration–time profile within the timeframe of study (see dotted black curve in Fig. 4a), but its $t_{1/2}$ may be comparable to the value for TMS25 (i.v.).

chorioallantoic membrane model, which recapitulates the diverging–converging capillary vasculature associated with sink-like organs such as the liver and spleen. It was shown that PEI50 rapidly binds to serum proteins and endothelial cells in comparison to QA50. The subtle difference in surface chemistry arguably alters the vulnerability of PEI50 MSNs to phagocytosis because of increased opsonization and hence reduced systemic residence and is consistent with previous studies of the effects of surface chemistry on protein corona[42,43].

Interestingly, we observed no significant effect of zeta potential on the AUC$_{0-24\,h}$ of size-matched and surface chemistry-matched, but differently charged TMS50 and QA50 particles (unpaired $t$ test, $P > 0.05$) (see Fig. 4c), although the positively charged QA50 has a slightly lesser $k_{out}$ (hence slightly greater $t_{1/2}$) than neutral TMS50 (unpaired $t$ test, $P < 0.05$) (see Fig. 4e and Table 2). As seen before in Fig. 2b, h, the washout of TMS50 from thorax is accompanied by increased concentration of MSNs in the liver and spleen, but that of QA50 is accompanied primarily by excretion into the large intestine and urinary bladder, indicating that the positively charged particles tend to be excreted out faster than their neutral counterparts, which tend to be sequestered in the liver and spleen longer[12]. This difference, however, does not cause variation in the systemic bioavailability of the two particles as indicated by similar values of AUC$_{0-24\,h}$.

**Individual-organ kinetics.** As described before, we theoretically classify ROIs in the body as source-like and sink-like, which becomes more evident as we consider the kinetic behavior of MSNs in individual ROIs. Because source-like ROIs, namely

lungs, abdominal aorta, muscles, and brain, are deficient in traps (e.g., fenestrations are not large enough to allow transvascular escape of NPs[34,35]), MSNs are not sequestered into the interstitium and only traverse through the blood pool of these ROIs. Hence, a mono-exponential decay function (Eq. (6)) explains the concentration–time course of MSNs through such ROIs (see Fig. 5a, d, Supplementary Figs. 12a, b, 13a, b, and 14a, b, and Supplementary Tables 4 and 5). MSNs demonstrate synchronous behavior across all source-like ROIs, which in turn closely resembles their behavior to systemic kinetics (i.e., heart ROI) (Fig. 4a, b). A change in the heart concentration of MSNs is reflected by a similar change in source-like ROIs, as is also evident from similarity in the $k_{out}$ values of MSNs across ROIs (one-way ANOVA, $P > 0.05$), except PEI50 (one-way ANOVA, $P < 0.05$) (comparing Fig. 4e and Supplementary Figs. 8c, 12f, 13g, and 14g). This is strongly suggestive of coupling between the heart and source-like ROIs; heuristically, the underlying reason lies in the similar microanatomy of these ROIs. As seen in the ROI to heart concentration ratios (Fig. 5g and Supplementary Figs. 12c, 13c, and 14c), source-like ROIs have an almost constant ratio over time, which corroborates the coupling of source-like ROIs to the heart. Also, the mathematical relation $AUC_{0-24\,h} = \lambda \cdot size^{-n}$ seems to hold true for all source-like ROIs, with values of power coefficient $n$ ranging from 0.5 to 0.9, suggesting a moderate to strongly negative linear dependence (Fig. 5j, Supplementary Figs. 12e, 13e, and 14e, and 19a, b, c, f). All of the above suggest that the effect of MSN physicochemical properties and routes of administration on MSN disposition kinetics in source-like ROIs is similar to that in systemic kinetics. The concentration levels, however, do vary across these ROIs because of differences in organ perfusion (see Fig. 3).

As discussed previously, the sink-like ROIs (i.e., liver, spleen, and thoracic lymph nodes) behave differently than the heart and source-like ROIs (primarily due to porous capillaries with discontinuous endothelium[25,34,35,44] and high density of macrophages[24,44–46]), and as a result we observe a varying ROI to heart concentration ratio over time (Fig. 5h, i and Supplementary Fig. 11b). MSN concentrations in these ROIs rise over time initially, followed by slow or no decline in concentration within 24 h, indicating the presence of traps causing MSN accumulation over time into the interstitium (see Fig. 5b, c, e, f and Supplementary Fig. 11a). As a result in sink-like ROIs (Fig. 5b, c, e, f and Supplementary Fig. 11a), Eq. (4) or its adaptation, Eq. (5), is fit to MSN concentration–time data (see Supplementary Tables 4 and 5).

As to the effect of MSN size, a larger size of TMS-coated MSNs is associated with a greater $AUC_{0-24\,h}$ in sink-like ROIs, irrespective of the route of administration, indicating a greater accumulation of MSNs (see Fig. 5k, l and Supplementary Fig. 11d). The mathematical relation between $AUC_{0-24\,h}$ and particle size, $AUC_{0-24\,h} = \lambda \cdot size^{n}$, is consistent across all sink-like ROIs with values of power coefficient $n$ varying between 0.4 and 1.2, suggesting a moderate to strongly positive linear dependence (see Supplementary Fig. 19d, g, h). Comparing the effect of route of administration, i.v. delivered TMS-coated MSNs are associated with a higher $AUC_{0-24\,h}$ value than their i. p. delivered counterparts in the spleen (unpaired $t$ test, $P < 0.05$), but with comparable values in the liver (unpaired $t$ test, $P > 0.05$) (see Fig. 5k, l).

For the PEI50 MSNs with surface-exposed amines, spleen and liver are the prime sites of radioactivity, unlike the size-matched and zeta potential-matched counterpart, QA50, with obstructed amines (see Figs. 3a, 5e, f). QA50, however, shows resemblance in its behavior to TMS50, indicating that surface chemistry plays a more prominent role in affecting the hepatic and splenic accumulation of MSNs compared to zeta potential.

It is worth mentioning that TMS25 (i.v.) and QA50 (i.v.), show a decline in concentration over time in the liver, in contrast to the other MSNs; hence, the mono-exponential Eq. (6) was fit to their concentration–time course (Fig. 5b, e). The difference is also evident from the almost constant liver-to-heart concentration ratio for these two MSNs, unlike other MSNs which exhibit an increasing ratio over time (Fig. 5h). The small size of TMS25 seems unfavorable for MSN sequestration in the liver[9] and the positive zeta potential of QA50 seems favorable for hepatobiliary elimination[12], consistent with published literature, hence overall a low sequestration in the liver is observed leading to an almost constant liver to heart concentration ratio. This information is valuable for MSN design optimization.

**Excretion kinetics.** Urine and feces were not collected during the in vivo study; we thus examine kidneys, urinary bladder, and total excreted activity data (Fig. 6a–f) to understand the excretion kinetics of MSNs. A mono-exponential decay function (Eq. (6)) was fit to the concentration–time data in the kidneys following i.v. injection, and Eq. (4) was fit to the data obtained following i.p. injection. As seen in Fig. 6a, d, there is a tight overlap between the kidney concentration–time profiles of various MSNs injected i.v. or i.p., with a mean radioactivity concentration of <2.5%ID g$^{-1}$ at 30 min in all cases, and an overall trend of decline in concentration over time. Because the radioactivity in kidneys is a combined result of $^{111}$In-DTPA (indium-diethylenetriamine pentaacetic acid (DTPA) chelate) present in the glomerulus and collecting duct system, we instead refer to the urinary bladder ROI and the total excreted activity data for an understanding of the excretion behavior of MSNs.

The presence of radioactivity in the urinary bladder ROI indicates that some $^{111}$In-DTPA is reaching the bladder (Figs. 2, 3, 6b, e). We note that the bladder signal is unlikely to arise from extravasated NPs in the bladder wall, given the uniformity of radioactivity across the bladder volume evident in thin (0.4 mm) sections (Supplementary Fig. 17) and the observation that other smooth muscle structures (such as the trachea, stomach, and intestines) show no significant uptake. Thus, the activity in the bladder arises from one of the following pathways: (a) glomerular filtration of intact NPs through kidneys[47], or (b) renal excretion of free $^{111}$In-DTPA following degradation of NPs in circulation[9]. Given the high stability of the $^{111}$In-DTPA label on MSNs in physiologically relevant media (Supplementary Fig. 5), possibility "b" diminishes. Thus, pathway "a" seems to be the primary mechanism responsible for visible radioactivity in the bladder. This observation defies the often-quoted renal clearance cutoff of ~5.5 nm[5]. Although there are published studies[13,48,49] that demonstrate excretion of intact NPs as large as 110 nm in urine, and an in-house in vivo experiment that shows urinary excretion of intact 50 nm PEGylated TMS MSNs in mice (Supplementary Fig. 18), future investigations involving microscopic examination of urine samples will be necessary to validate this hypothesis and to gain an understanding of the mechanism of excretion. Due to loss of activity from the bladder via urination, the activity detected in the bladder at a given time point (Fig. 6b, e) is not equivalent to the cumulative amount of renal excretion. The urinary bladder activity thus gives an incomplete picture of renal excretion. Therefore, we quantify total excretion kinetics instead, which accounts for urinary and fecal excretion combined, to analyze excretion kinetics of MSNs. We fit Eq. (7) to the total excreted activity (%ID) over time, which is obtained by subtracting the measured whole-body activity (%ID) at any time $t$ from the

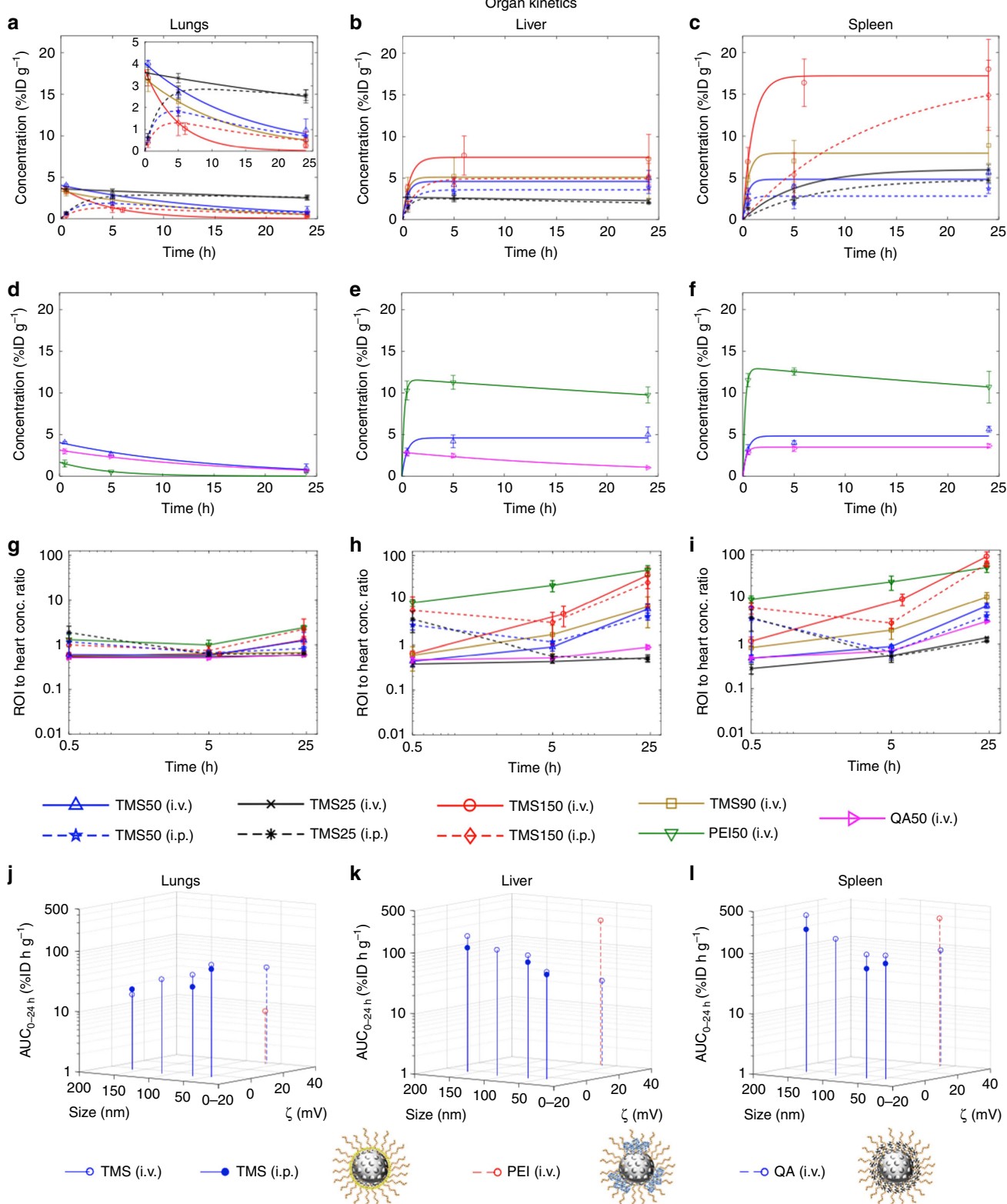

**Fig. 5** Organ kinetics. Panel shows kinetic analysis for lungs, liver, and spleen. For analysis of the remaining ROIs refer to Supplementary Figs. 11–15. **a–f** For each ROI, non-linear regression of Eq. (4) or its adaptation was performed to the concentration–time data (Supplementary Tables 4 and 5). Fitted concentration–time curves demonstrate the effect of MSN size and route of administration for TMS-modified MSNs (**a–c**) and surface chemistry and zeta potential for 50 nm diameter MSNs modified with TMS, QA, or PEI (**d–f**). The inset in **a** is a rescaled version of the figure for a clearer view. Solid lines, i.v. cases; dotted lines, i.p. cases. **g–i** Observed concentration of ROIs normalized to concentration of heart (substitute for plasma) is shown over time on a log–log plot. **j–l** 3-D stem plots show area under the concentration–time curves ($AUC_{0–24 h}$), obtained for different MSNs from **a–f**, in multiparameter space. Data represent mean ± s.d., $n = 4$ (except TMS50 (i.p.) and TMS25 (i.p.), where $n = 3$)

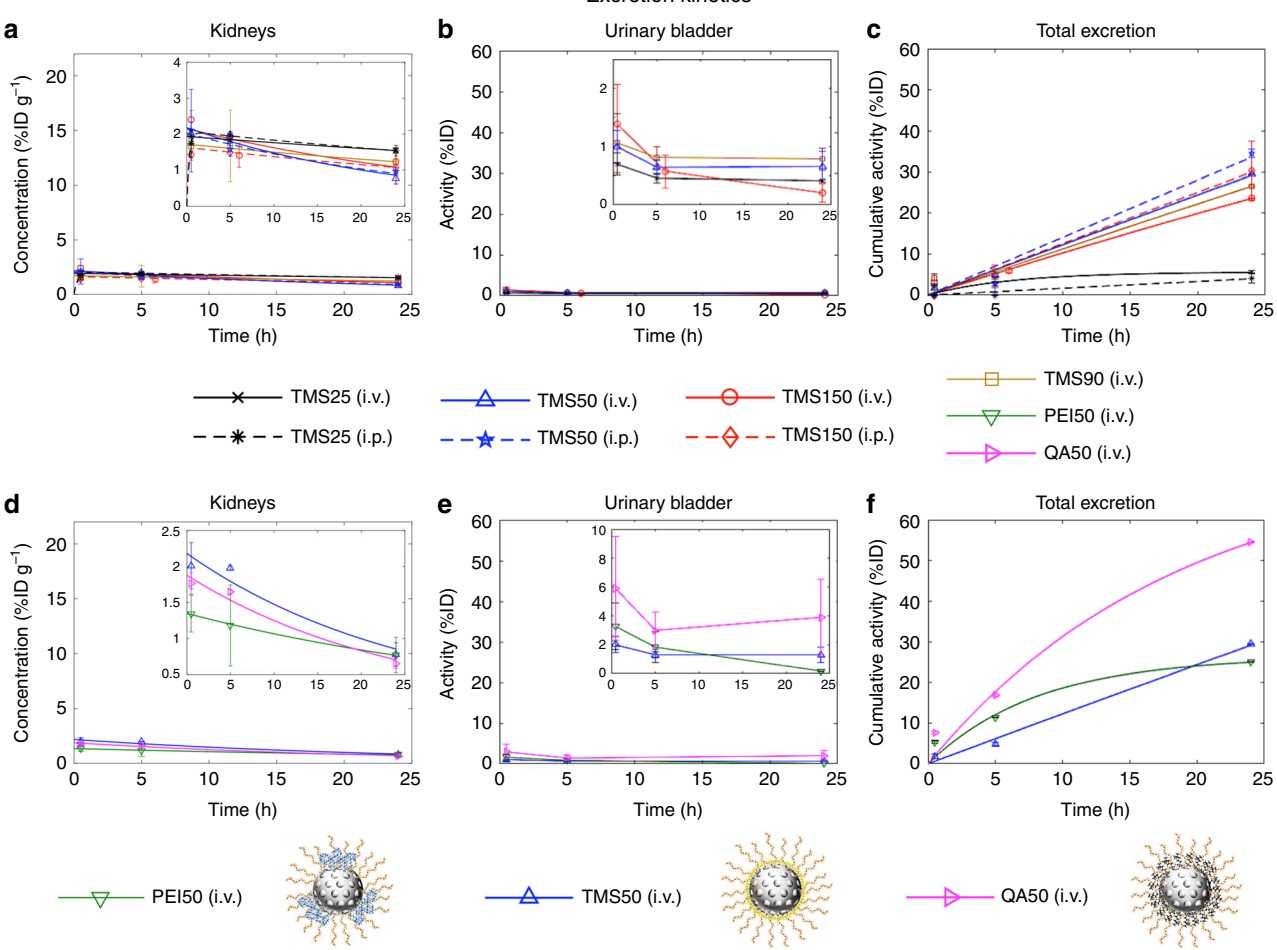

**Fig. 6** Excretion kinetics. Urine and feces were not collected during the in vivo study, we thus examine the kidneys, urinary bladder, and total excreted activity to infer excretion kinetics of MSNs. **a**, **d** Concentration–time data of kidneys was fit to Eq. (6) for i.v. cases and Eq. (4) for i.p. cases. **b**, **e** No model was fit to the urinary bladder activity data due to the lack of urine collection. Data are presented as activity in the bladder as a percentage of the injected dose (%ID) over time. **c**, **f** To the total excreted activity (100%ID—whole-body activity) data, Eq. (7) was fit to all cases (Supplementary Tables 4 and 5). The insets in **a**, **b**, **d**, and **e** are rescaled versions of their corresponding figures for a clearer view. Solid lines, i.v. cases; dotted lines, i.p. cases. Data represent mean ± s.d., $n = 4$ (except TMS50 (i.p.) and TMS25 (i.p.), where $n = 3$)

measured activity of the injected dose (100%) (Fig. 6c, f and Supplementary Tables 4, 5).

As can be seen in Fig. 6c, TMS-coated MSNs of different sizes injected i.v. or i.p. are excreted to a comparable amount (except TMS25, which has the least amount excreted). Otherwise, there is no significant correlation between size or route of administration and excretion. Regarding charge, in a previous study, substantial excretion was observed in <30 min with positively charged particles, but was much slower (up to several days) with strongly negatively charged particles[12]. In our study, QA50, which is strongly positively charged, shows higher excretion than neutral MSNs (Fig. 6f), consistent with the previous study. However, the other highly positively charged MSN used in our study, PEI50, shows lesser excretion than QA50 within 24 h, apparently because sequestration by the liver and spleen delays excretion (see Figs. 2i, 6f). The higher total excretion of QA50 correlates with lower MPS organ accumulation, higher bladder activity, and higher large intestine activity (see Fig. 2h), which suggests that QA50 tends to have higher urinary and fecal excretion, compared to the other MSNs.

## Discussion

We have demonstrated the application of a combined mathematical modeling and non-invasive SPECT/CT imaging approach to PK analysis of MSNs. The selection of MSNs as the NP of choice for the current investigation is based on their ability to undergo surface functionalization and precise synthesis control that allows for selection of particle size, shape, and pore size. Furthermore, the safety of MSNs is supported by the fact that amorphous silica is generally recognized as safe as a food additive by the FDA, and recently amorphous silica NP "C dots" (Cornell dots) were FDA approved for diagnostic applications in a phase I clinical trial[50], making MSNs ideal candidates for drug-delivery systems[51–54]. The range of particles used to test the effect of size, charge, and surface chemistry reveals that the in vivo biodistribution and clearance of MSNs is significantly affected by their physiochemical properties. We justified the classification of ROIs into source-like and sink-like organs based on their underlying physiological differences and observed NP kinetics, and applied semi-mechanistic models to the concentration–time profiles of NPs in these ROIs to determine relevant PK parameters. Our analysis showed that smaller MSN size results in a higher systemic bioavailability, irrespective of the route of administration; positive charge favors greater excretion; and importantly, surface-exposed charged molecules (amines) increase vulnerability to sequestration in the liver and

spleen. Notably, a consistent mathematical relation between one key PK parameter ($AUC_{0-24\,h}$) and MSN core diameter was identified in the form of $AUC_{0-24\,h} = \lambda \cdot size^{-n}$ for systemic circulation and all source-like organs in both i.v. and i.p. cases; however, for sink-like organs, the relation identified is $AUC_{0-24\,h} = \lambda \cdot size^{n}$.

Regarding the predictive power of our semi-mechanistic mathematical model, it operates at a macroscopic scale, meaning that it is based on the organ-scale or tissue-scale concentration–time profiles of NPs wherein several microscopic mechanisms are lumped together into the phenomenological macroscopic constants and variables used in the model. The predictive capacity of the model is thus limited in scale. The model can help to reliably predict organ exposure of NPs based on functional relationships (vide supra) between organ exposure ($AUC_{0-24\,h}$) and MSN size through interpolation within the studied size range (~32 to ~142 nm). As for extrapolation beyond the studied size range, we expect its accuracy to worsen for NP size below 5.5 nm. Because of predominance of renal clearance below 5.5 nm, systemic circulation and exposure of source-like and sink-like organs to such particles will be drastically reduced, which means that the correlation function for at least systemic circulation and source-like organs, if not for sink-like organs, will be reversed as per a published report[5]. Thus, we would like to assume 5.5 nm as the safe lower bound for the functions defined in our study. For sizes above 142 nm, we expect the functional relationship discovered here to generally apply, as larger sizes of NPs should continue to correlate with even greater hepatic and splenic uptakes[55]. Further, as for the relationship between $AUC_{0-24\,h}$ and zeta potential, we find no significant effect of zeta potential on the systemic circulation and source-like organ exposure of MSNs. However, positive charge with shielded surface amines (QA) correlates with greater total excretion, hence lower liver accumulation compared to neutral MSNs (TMS) or cationic MSNs with surface-exposed amines (PEI). The same trend holds true for possible urinary excretion and seems reasonable given the presence of anionic charge within the glomerular capillary wall[7], but most importantly it highlights the importance of surface exposure of charged molecules in affecting in vivo interactions. Based on these trends and on previously published studies[12,28], we can extrapolate that anionic MSNs will have reduced hepatobiliary and urinary excretion, which needs further investigation for conclusive evidence. Furthermore, based on the scope of our study, we propose 32-nm TMS-alkylated MSNs and 56-nm QA-aminated MSNs for therapeutic applications, primarily due to their low hepatic and splenic accumulation. Since 32-nm TMS-alkylated MSNs stay in circulation nearly four times longer than 56-nm QA-aminated MSNs, in applications demanding longer circulation times, for example, tumor delivery, the neutral alkylated MSNs appear to be a better choice over the positive aminated MSNs.

## Methods

**Overview**. Bolus tail vein (i.v.) or i.p. injection of MSNs conjugated with radioactive [111]In was given to healthy female rats, followed by whole-body SPECT/CT imaging of animals longitudinally over 24 h. ROI (Fig. 1c) analysis was performed on reconstructed SPECT/CT images to obtain dose normalized radioactivity concentration–time-course data (Fig. 3). Semi-mechanistic modeling and PK analyses were then performed to understand the effect of physicochemical properties and routes of administration on MSN disposition kinetics. To study the effect of MSN size and route of administration, PEG-TMS-coated MSNs of four different nominal sizes (25, 50, 90, and 150 nm) were administered i.v., and three different nominal sizes (25, 50, and 150 nm) were administered i.p. Further, size-matched and surface chemistry-matched MSNs (TMS50 and QA50) were used to study the effect of zeta potential. Finally, to study the effect of surface chemistry, size-matched and zeta potential-matched particles (QA50 and PEI50) were used.

**NP synthesis and characterization**. The synthesis of colloidally stable PEGylated MSNs with various sizes and different surface chemistries was based on published methods[16–18]. To enable detection by SPECT, monodisperse MSNs were covalently coupled to DTPA through isothiocyanate and amine reactions to enable binding of [111]In, a gamma-emitting radioisotope with a radioactive half-life of 2.8 days[56]. First, 7.5 mg of S-2-(4-isothiocyanatobenzyl)-diethylenetriamine pentaacetic acid (p-SCN-Bn-DTPA, Macrocyclics, Pano, TX, USA), 3.75 μL of 3-aminopropyltriethoxysilane (APTES, Sigma-Aldrich, St. Louis, MO, USA), and 15 μL of trimethylamine (Sigma-Aldrich, St. Louis, MO, USA) were mixed in 1 mL of anhydrous ethanol under continuous agitation for 18 h. Then, 0.29 g of cationic surfactant, n-cetyltrimethylammonium bromide (CTAB, Sigma-Aldrich, St. Louis, MO, USA) was dissolved in 150 mL of ammonium hydroxide (NH₄OH, Sigma-Aldrich, St. Louis, MO, USA) solution and heated to 50 °C. After 1 h, dilute tetraethyl orthosilicate (Sigma-Aldrich, St. Louis, MO, USA) solution (prepared in ethanol) and APTES/p-SCN-Bn-DTPA mixture solution were added simultaneously to the CTAB containing ammonium hydroxide solution. After an additional 1 h of continuous stirring, 2-methoxypolyethyleneoxy-propyltrimethoxysilane (Gelest, Morrisville, PA, USA) was added to the solution and the mixture was stirred for 30 min, and then a secondary silane (trimethylsilane, TMS, Sigma-Aldrich, St. Louis, MO or trimethoxysilylpropyl modified polyethyleneimine, 50% in isopropanol, MW 1500–1800, PEI-silane, Gelest, Morrisville, PA, USA or N-trimethoxysilylpropyl-N,N,N-trimethyl ammonium chloride, 50% in methanol, TMAC-silane) was added. Stirring was stopped after an additional 30 min, and the solution was stored at 50 °C for 20 h. Solutions were then sealed and stored at 90 °C for 24 h for hydrothermal treatment. Next, we followed a procedure for CTAB extraction described previously in the literature[57]. Prior to use, MSNs were transferred to deionized water at a concentration of 10 mg mL⁻¹.

The detailed conditions and amounts of chemical reagents used in the preparation of PEG-TMS-modified, PEG-PEI-modified, and PEG-QA-modified MSNs are described in Supplementary Table 1.

To label MSNs with [111]In, a solution of InCl₃ (Inidiclor, GE Healthcare, Arlington Heights, IL, USA) was incubated with the DTPA-modified PEG-TMS, PEG-PEI, or PEG-QA NPs, using 15 mCi of In-111 per 10 mg particles, for 30 min at room temperature in 500 mM sodium citrate buffer. Unbound [111]In was removed by centrifugation at $21,000 \times g$ for 60 min, followed by resuspension in 1 mL of 1× PBS at 10 mg mL⁻¹. No loss of radioactivity from the MSNs was observed following two subsequent washes.

The purified PEGylated MSNs were characterized by TEM, DLS, and zeta potential prior in vivo injections. TEM images were acquired with a JEOL 2010 (200 kV voltage, Tokyo, Japan) instrument equipped with a Gatan Orius digital camera system (Warrendale, PA, USA). Hydrodynamic size and zeta potential analyses were performed on a Malvern Zetasizer Nano-ZS equipped with a He-Ne laser (633 nm) and non-invasive backscatter optics. All samples for DLS or zeta potential measurements were suspended in either PBS or 10 mM NaCl at 200 μg mL⁻¹. Measurements were acquired at 25 °C in triplicate. The Z-average diameter and number particle size distribution was used for all reported hydrodynamic diameter measurements. The zeta potential for each sample was obtained from monomodal analysis measurements.

Further, to characterize the textural properties of MSNs, bare MSN samples were degassed at 60 °C for 16 h under vacuum prior to measuring N₂ sorption at 77 K on an Autosorb iQ2 (Quantachrome, Boynton Beach, FL, USA). Pore size distributions were modeled using Non-Local Density Functional Theory assuming a silica surface and cylindrical pores. Surface areas were determined using the Brunauer–Emmett–Teller method over the relative pressure range $P/P_0 = 0.05$ to 0.15.

**Stability testing of Indium-labeling of MSNs**. To test the stability of In labeling of MSNs, we exposed In-labeled MSNs to various physiologically relevant media: acetate saline, PBS, and SBF (Supplementary Table 3). MSNs of nominal size (50 nm) and all three surface chemistries (PEG-TMS, PEG-QA, and PEG-PEI) were studied. Freshly made non-radioactive In-labeled MSNs were suspended in the three buffers (at 1 mg mL⁻¹) and incubated in the dark at 37 °C under gentle shaking for 4 and 48 h. The suspension was then centrifuged (30 min, 21,000 rcf), and the isolated pellet was resuspended in distilled water (5 mL). This procedure was repeated twice. All supernatants and MSNs were used for elemental analysis of In content, as detailed below.

Supernatant samples after exposure to In-labeled MSNs were analyzed as-received using graphite furnace atomic absorption spectrophotometry (Perkin Elmer PinAAcle 900T, USA). A matrix modifier containing Pd and Mg(NO₃)₂ was used for all standards, blanks, and samples. Commercial InCl₃ (Aldrich) was used for preparing standard solutions.

Solid In-labeled samples (after leaching) were dispersed in HCl and repeatedly washed to extract indium into solution. Solutions were further diluted with deionized water prior to analysis via flame atomic absorption spectrophotometry (Perkin Elmer PinAAcle 900T, USA). To verify that the acid leaching effectively extracted all indium from the samples, a second analysis was conducted using a mixture of HCl and HF to ensure digestion of all solid material. Analytical results were the same for both sets of analyses.

**Animal study design, SPECT/CT imaging, and quantification.** All procedures involving rats were conducted in accordance with the National Institutes of Health regulations concerning the care and use of experimental animals. This study was approved by the University of New Mexico Health Sciences Center Institutional Animal Care and Use Committee (protocol #13-101096-HSC) and the USAMRMC Animal Care and Use Review Office (protocol #CB-2013-29.03).

Healthy female Fischer 344 rats (approx. 150 g each) were used in these studies. Each rat was administered 1 mg of particles, suspended in 200 μL of 0.5× PBS, and labeled with approximately 1 mCi of $^{111}$In by either tail vein (i.v.) or i.p. injection. For each subject, the activity of the injected dose was measured immediately prior to injection to enable appropriate normalization of quantitative imaging results for each subject. Four groups ($n = 4$ rats per group) were administered PEG-TMS-coated particles of different nominal sizes (25, 50, 90, or 150 nm) by tail vein injection. Two additional groups ($n = 4$ rats per group) received 50 nm particles i.v. coated with PEG-PEI or PEG-QA, respectively. Further, three additional groups were administered 1 mg of PEG-TMS particles of different nominal sizes (25, 50, or 150 nm) by i.p. injection ($n = 3$ rats per group, except TMS150, with $n = 4$ rats per group).

Advancements in small-animal imaging techniques[58,59] have enabled whole-body, three-dimensional, dynamic imaging in rodents to quantify biodistribution of radiolabeled xenobiotics in the presence of an anatomical reference. These techniques provide the ability to study spatio-temporal evolution of whole-body biodistribution non-invasively within the same animal, presenting a significant advantage over blood sampling and organ resection. SPECT/CT imaging[59] was conducted at the Keck-UNM Small Animal Imaging Resource using a dual-modality NanoSPECT/CT® Small Animal In Vivo Imager (Bioscan, Inc., Washington, DC, USA). For each subject, the $^{111}$In biodistribution was imaged longitudinally at three time points (30 min, 5 h (6 h instead for TMS150 (i.v.) group), and 24 h) post injection with the rat maintained under isoflurane anesthesia on a heated bed (37 °C) during imaging. The computed tomography (CT) acquisition (approx. 5 min duration) was completed using 180 projections with a pitch of 1.5. Helical SPECT acquisition included 32 projections and varying time per projection resulting in an acquisition time of 15–30 min per time point. Immediately after the 24 h imaging time point, each rat was euthanized, and tissues were harvested and fixed in 10% formaldehyde for future analysis by microscopy.

The SPECT/CT image data were exported to the VivoQuant 2.00 software (inviCRO, LLC, Boston, MA, USA) for image reconstruction, display, and analysis. Camera calibration and reconstructions were performed using both $^{111}$In gamma energy windows (0.1713 and 0.2454 MeV). Co-registered CT and SPECT axial images were reconstructed with a 176 × 176 matrix, 0.4 mm in-plane resolution, and a slice thickness of 0.4 mm. The number of slices for each whole-body image was approximately 450. Tissue segmentation and ROI analysis were performed by inviCRO, LLC. ROIs corresponding to the whole body, brain, liver, kidneys, spleen, heart, lungs, lymph nodes, bladder, abdominal aorta, bone (knee joint), and muscles (Fig. 1c) were selected according to the following procedure: except for muscle and bone ROIs, which were generated manually, ROIs were generated using inviCRO's Multi Atlas Segmentation Tool. First, fixed volume ROIs were placed manually for 10 CTs, and used as a reference library for 10 additional scans. The final reference library included all 20 CTs. The reference CTs were registered to each new data set using both affine and deformable registration. The reference ROI had the same transform applied to it, resulting in 20 representations of possible ROI locations. Finally, using the best five registrations, a probability map of each ROI was created and thresholded to generate a final ROI of the correct volume. At each time point, ROIs were quantitatively analyzed to determine the decay-corrected activity normalized by the activity of the injected dose (expressed as %ID g$^{-1}$) based on the total activity detected in the ROI, the ROI volume, and the tissue density.

It is important to note that in situations where radioactivity quantification of an organ can potentially be confounded by anatomically adjacent organs of interest, ROI analysis is performed on a section of the organ, instead of using the entire organ. This approach assumes homogeneity of concentration across the organ. For example, as can be seen in Fig. 1c, lungs and liver are analyzed based on sections defined within the organs to avoid overlap with the heart and spleen, respectively. This strategy limits the partial volume artifacts.

**Semi-mechanistic mathematical modeling and PK analysis.** Different from our prior work on modeling free drug[60–66] and targeted nanocarrier delivery to tumors[67–70], we here used a parsimonious, semi-mechanistic model to describe the macroscopic concentration–time behavior of MSNs in individual "black box-like" ROIs and estimate relevant PK parameters. As shown in Fig. 1b, an organ $i$ receives an influx of NPs from the major feeding artery. Based on the characteristics of NPs and the organ anatomy and physiology at the microvascular scale, NPs traverse through the vasculature of organ $i$ while forming transient or permanent associations with intravascular traps. The untrapped fraction of incoming particles is free to leave organ $i$ and rejoin the venous blood. These interactions at the microvascular scale thus govern the global biodistribution profile of NPs. Given the nature of data in the current study, we do not model NP interactions at microscopic scale, but only phenomenologically describe the observed macroscopic concentration–time behavior of NPs using a parsimonious model.

Our model is based on the hypothesis that superposition of two opposing first-order processes of influx and efflux of NPs, through the vasculature of an ROI, can explain the observed concentration–time course of NPs in the given ROI (see Fig. 1b). We thus obtain the following differential equation describing the rate of change of concentration $C_i$ (units, %ID g$^{-1}$) of NPs in organ $i$:

$$\frac{\mathrm{d}C_i}{\mathrm{d}t} = k_{\mathrm{in},i} \cdot C_{\mathrm{b},i} - k_{\mathrm{out},i} \cdot C_i, \tag{1}$$

where $k_{\mathrm{in},i}$ and $k_{\mathrm{out},i}$ are the first-order uptake and elimination rate constants, respectively (units, h$^{-1}$); and $C_{\mathrm{b},i}$ is the concentration of NPs in the local arterial blood supply of organ $i$, which changes at a rate assumed to be governed by a first-order disposition process:

$$C_{\mathrm{b},i}(t) = C_0 \cdot e^{-k_{\mathrm{in},i} \cdot t}. \tag{2}$$

Here, $k_{\mathrm{in},i}$ is the first-order rate at which NPs are being supplied by the artery to the ROI; $C_0$ is the concentration of NPs at time $t = 0$.

For the i.v. bolus case, $C_0$ is achieved immediately after injection, but for the i.p. case, because absorption of NPs form peritoneum into blood circulation is a time-dependent process, for simplification we assume that $C_0 = C_{\mathrm{avg}}$, where $C_{\mathrm{avg}}$ represents the average concentration of NPs due to absorption alone in systemic circulation during the period when NPs are being absorbed from peritoneum into systemic circulation. It can be expressed as:

$$C_{\mathrm{avg}} = \frac{\int_0^t C_{\mathrm{b}} \mathrm{d}t}{t}. \tag{3}$$

We further assume that the rate of absorption is constant, resulting in $C_{\mathrm{avg}} = C_{\mathrm{b}}^{\max}/2$, which is mathematically equivalent to the zeroth-order approximation of a Taylor series.

The integrated form of Eq. (1), solved for initial condition $C_i(0) = 0$ is:

$$C_i(t) = A \cdot \left(e^{-k_{\mathrm{out},i} \cdot t} - e^{-k_{\mathrm{in},i} \cdot t}\right), \tag{4}$$

where the macro-constant $A = \frac{k_{\mathrm{in},i} \cdot C_0}{k_{\mathrm{in},i} - k_{\mathrm{out},i}}$, is the intercept of back-extrapolated elimination phase of double-exponential concentration–time curve of an ROI. Based on empirical evidence from the quantified SPECT/CT images, Eq. (4) was further adapted to model the behavior of individual source-like and sink-like ROIs, under i.v. or i.p. conditions of MSN administration.

Following i.p. delivery, for all source organs and lymph nodes (sink-like organ), an obvious uptake phase followed by an elimination phase is observed; thus, we fit Eq. (4) in its canonical form to the concentration–time data. However, for the remaining sink-like organs following i.p. delivery, an apparent elimination phase was not seen within the duration of study, thus assuming $k_{\mathrm{out}} \cong 0$, Eq. (4) becomes:

$$C_i(t) = A \cdot \left(1 - e^{-k_{\mathrm{in},i} \cdot t}\right). \tag{5}$$

Further, fitting Eq. (4) to the source-like organs in i.v. case, we found that $k_{\mathrm{in}} \gg k_{\mathrm{out}}$, and as a result, the second exponential term in Eq. (4) becomes insignificant, reducing the equation for source-like ROIs to:

$$C_i(t) = A \cdot e^{-k_{\mathrm{out},i} \cdot t}. \tag{6}$$

And, in the case of sink-like organs following i.v. injection, Eq. (4) is employed to model the empirical behavior of MSNs in the liver, and Eq. (5) in the spleen, based on whether or not elimination is seen in the data. For the total excreted activity obtained by subtracting the whole-body activity (%ID) at any time $t$ from 100% activity, we use an adaptation of Eq. (5):

$$U(t) = U_t \cdot \left(1 - e^{-k_{\mathrm{u}} \cdot t}\right), \tag{7}$$

where $U$ is the total excreted activity at time $t$ (units %ID), $k_{\mathrm{u}}$ is the first-order excretion rate constant (units h$^{-1}$), and $U_t$ is the total amount of MSNs excreted (units %ID). Total excreted activity accounts for both renal and hepatobiliary excretion.

To estimate model parameters and correlate them to the physiological and physicochemical underpinnings of the observed in vivo behavior of MSNs, we performed non-linear regression analyses of the semi-mechanistic models (Eqs. (4)–(7)) to concentration (or, cumulative activity)–time data of individual ROIs. Further, we performed traditional PK analysis by employing the concentration–time data of the heart ROI as a substitute for plasma concentration–time-course data. This substitution assumes that radioactivity from the heart ROI is purely due to NPs in the blood pool of the heart, and not in the extravascular tissue space (based on our previous discussion of the heart being a source organ). It could be argued that the data from a blood vessel ROI should be used as a surrogate for plasma, but we have used the heart instead, because the reliability of the segmentation of the heart is much greater than that of the blood vessels. Even the aorta is physically too small (~1 mm diameter), relative to the resolution of the SPECT data (0.4 mm), to accurately segment the vessel lumen.

Based on the nature of concentration–time curves of MSNs in the heart, we applied a one-compartment PK model[36] (same as Eq. (6) for i.v. delivery, and Eq. (4) for i.p. delivery), and determined PK parameters: (i) AUC from 0 to 24 h

(AUC$_{0-24\,h}$), (ii) uptake rate constant ($k_{in}$), (iii) elimination rate constant ($k_{out}$), and (iv) half-life ($t_{1/2}$). AUC$_{0-24\,h}$ represents the systemic bioavailability of NPs and is the definite integral of NP concentration–time in plasma (heart, in this study), determined analytically. $k_{out}$ is the slope of the curve on a semi-log plot between $C_{heart}$ and $t$, and represents the fraction of NPs eliminated from plasma per unit time. $t_{1/2}$ is the time required for NP concentration to reduce to half, and for a one-compartment model, $t_{1/2}$[36] is obtained as:

$$t_{1/2} = \ln(2)/k_{out}. \tag{8}$$

Further, we also estimated model parameters for all the other ROIs to understand the effect of MSN characteristics and route of administration on organ exposure to MSNs, and their uptake and elimination behaviors.

**Statistical analysis**. For in vivo studies, four animals per group were used. One subject from the TMS25 (i.p.) group and one subject from the TMS50 (i.p.) group were excluded from analysis due to subject motion and a misplaced injection, respectively, resulting in $n = 3$ for these groups. Experimental results are presented as mean ± standard deviation (s.d.). One-way ANOVA and Tukey's honest significant difference procedures were performed to evaluate differences in model parameters across groups. Unpaired-sample $t$ test was also performed for relevant pairwise comparisons. $P < 0.05$ was considered statistically significant. The "Levenberg–Marquardt" algorithm was used to perform non-linear regression analysis to the observed data. All analyses were performed in MATLAB R2015b.

## Data availability

The data that support the plots within this paper and other findings of this study are available from the corresponding authors on reasonable request.

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

## Acknowledgements

Nanoparticle synthesis and SPECT imaging studies were supported by the Defense Threat Reduction Agency (DTRA) Nanostructured Active Therapeutic Vehicles (NATV) Program through contracts DTRA1002713506 and DTRA100279003 awarded to Sandia National Laboratories. C.J.B. and K.S.B. acknowledge additional support from the Sandia National Laboratories Laboratory Directed Research and Development Program. We also acknowledge T.A. Daniels and M. Nysus from the Keck-UNM Small Animal Radio-imaging (KUSAIR) for technical assistance with in vivo experiments and T.J. Liguori from inviCRO, LLC, Boston, MA for assistance with image analysis. P.D. acknowledges B.S. Wilson, R. Pasqualini, W. Arap, A. Dobroff, Y.L. Chuang, T. Brocato, B.M. Wheeler, J.D. Butner, R. Serda, J.O. Agola, D. Gupta, B. Mulligan, S. Montoya, C. De Souza, and H. Hui for helpful scientific discussions, and the New Mexico SpatioTemporal Modeling Center (STMC) Graduate Student Fellowship (NIH P50GM085273). Computational research was supported in part by the National Science Foundation Grants DMS-1562068 (Z.W., V.C.), DMS-1716737 (Z.W., V.C.); the National Institutes of Health (NIH) Grants 1U01CA196403 (Z.W., V.C.), 1U01CA213759 (Z.W., V.C.), 1R01CA226537 (Z.W., V.C., C.J.B.), 1R01CA222007 (Z.W., V.C.), U54CA210181 (Z.W., V.C.); the University of Texas System STARS Award (V.C.); and the Rochelle and Max Levit Chair in the Neurosciences (V.C.). Sandia National Laboratories is a multi-mission laboratory managed and operated by National Technology and Engineering Solutions of Sandia, LLC, a wholly owned subsidiary of Honeywell International, Inc., for the US Department of Energy's National Nuclear Security Administration under contract DE-NA0003525. This paper describes objective technical results and analysis. Any subjective views or opinions that might be expressed in the paper do not necessarily represent the views of the US Department of Energy or the United States Government.

## Author contributions

C.J.B. and V.C. conceived the research. N.L.A. performed the in vivo experiments with assistance from the KUSAIR staff. P.D. and N.L.A. performed image analysis with assistance from inviCRO. P.D., Z.W., and V.C. developed the model and performed model analysis. C.J.B., Y.-S.L., P.N.D., J.G.C., A.N., and E.N.C. synthesized and characterized nanoparticles. P.D., N.L.A., Z.W., Y.-S.L., K.S.B., P.N.D., E.L.B., V.C., and C.J.B. wrote the paper.

## Additional information

**Competing interests:** The authors declare no competing interests.

