## [Peer Review File · Nature Communications]

Reviewers' Comments:

Reviewer #1:

Remarks to the Author:

This is a second round of the review process with only few criticism remaining to be additionally addressed. The authors provided compelling arguments addressing all criticisms.

1. Overly simplistic PK model. This reviewer's opinion is that a PBPK model would be the best choice for data analysis. However, given authors reasons for missing key physiological parameter values relevant to NPs absorption and disposition, the presented model seems to be a realistic solution.
2. Poor experimental design. This reviewer's concern of model over parameterization is lessened given good agreement of reported parameter values with published ones. However, to address this criticism, a table reporting standard estimates and their standard errors in addition to Table S4 would be helpful. Table S4 reports means and SEMs of the few estimates for a given ROI. Such information is insufficient to statistically assess precision of parameter estimates, and indirectly, a potential model over parameterization.
3. NPs accumulation in the bladder. Authors comprehensively clarified this issue.

Response to referee

We thank the reviewer for his/her careful reading of our manuscript and for expressing his/her thoughtful comments, concerns, and suggestions. We address each comment on a point-by-point basis.

We have revised the manuscript, and added a new Supplementary Table 5 to address the reviewer's comment.

Reviewer #1 (Remarks to the Author):

This is a second round of the review process with only few criticism remaining to be additionally addressed. The authors provided compelling arguments addressing all criticisms.

1. Overly simplistic PK model. This reviewer's opinion is that a PBPK model would be the best choice for data analysis. However, given authors reasons for missing key physiological parameter values relevant to NPs absorption and disposition, the presented model seems to be a realistic solution.

Thank you.

2. Poor experimental design. This reviewer's concern of model over parameterization is lessened given good agreement of reported parameter values with published ones. However, to address this criticism, a table reporting standard estimates and their standard errors in addition to Table S4 would be helpful. Table S4 reports means and SEMs of the few estimates for a given ROI. Such information is insufficient to statistically assess precision of parameter estimates, and indirectly, a potential model over parameterization.

We thank the reviewer for their critical comment. We are providing the requested information in a new Supplementary Table 5.

3. NPs accumulation in the bladder. Authors comprehensively clarified this issue.

Thank you.